# Heat stress causes economic and welfare disparities across agroecological zones in Burkina Faso

Martial A. K. Houessou [1]✉, Zuhal Elnour[1,2], Qinqin Kong[3,4,5], Harald Grethe[1] & Matthew Huber [5]

Increased warming due to climate change can induce heat stress in humans and adversely affect labour productivity due to heat-related morbidity. Here, we use a simulation model to examine the effects of heat stress, through declined labour capacity under +1.5 °C and 3.5 °C warming scenarios on agriculture and welfare across the three agroecological zones (Sudanian, Sudano-Sahelian, and Sahelian) in Burkina Faso. In the two scenarios, domestic production declines, with outdoor labour-intensive sectors such as cropping and mining being the most affected, reducing gross domestic product by 9% and 20%, respectively. All households lose welfare in all scenarios except non-poor households in the +1.5 °C scenario. Across zones, crop production declines strongest in the crop-producing Sudanian and Sudano-Sahelian zones. In contrast, relative welfare losses are strongest for households in the Sahelian zone. The study highlights the most vulnerable sectors, household groups, and zones requiring urgent attention in heat stress adaptation and mitigation policies.

Since the mid-1970s, annual and seasonal average temperatures in the Sahara and Sahel have increased by 1 to 3 °C, representing the strongest warming globally[1]. Rising temperatures cause heat stress in humans, animals, and plants. Human heat stress occurs when the body receives heat more than it can tolerate without suffering physiological impairment[2]. It often results from a combination of environmental and metabolic heat modulated by clothing insulation[3].

Heat stress is associated with health risks such as heat exhaustion, heat stroke, and even death[4,5] Symptoms like fatigue, nausea, irritability, and muscle cramps characterize heat exhaustion[6,7]. In a more severe state, the body experiences heat stroke, which manifests through symptoms such as confusion, irrational behaviour and loss of consciousness[8]. Depending on the occupation, heat stress effects can be more severe for one individual than for another. Workers in sectors that involve more physical effort and take place primarily outdoors, such as agriculture, mining, and construction, are considered at higher risk than others[2,9].

Empirical evidence shows that heat stress reduces workers' productivity[2], which will negatively impact global GDP, with important variation across regions and sectors[10–13]. Developing regions, particularly sub-Saharan Africa and Southeast Asia, are expected to be disproportionately affected[10,12], with outdoor-intensive sectors suffering the greatest productivity losses[13]. A step-wise methodological approach is required to measure heat stress and model its economy-wide

implications. In previous studies, heat stress is frequently measured using the Wet Bulb Globe Temperature (WBGT)[14], based on which labour response functions were developed and employed to model heat stress-induced labour productivity loss[15–17]. This labour productivity loss is implemented through labour efficiency parameters in global computable general equilibrium (CGE) models, calibrated against Global Trade Analysis Project (GTAP) databases[10–13].

Important limitations and research gaps remain. While global and regional assessments are common, the heterogeneous effects of heat stress within individual countries have not been comprehensively explored. For example, Ref. 11 and Ref. 13 provide insights into specific countries—Germany and several West African countries, respectively—but do not focus on disparities within countries. Furthermore, they employed global models relying on the GTAP database with a base year up to 2014[18], which limits country-specific detail and recent events affecting the base economic state in the analyses.

This study fills the existing literature gaps through four key contributions. First, we conduct a country-specific analysis employing a comparative-static single-country CGE model and focusing on Burkina Faso. We chose Burkina Faso due to its severe vulnerability to climate change-induced heat stress. The country is located in Sahelian West Africa, one of the regions in the world most likely to see major increases in future heat stress, while simultaneously being especially vulnerable to

[1]International Agricultural Trade and Development Group, Humboldt-Universität zu Berlin, Berlin, Germany. [2]Agricultural Research Cooperation, Wad Madani, Sudan. [3]Department of Medicine, Stanford University, Stanford, CA, USA. [4]Woods Institute for the Environment, Stanford University, Stanford, CA, USA. [5]Department of Earth, Atmospheric and Planetary Sciences, Purdue University, West Lafayette, USAIN. ✉e-mail: martial.houessou@hu-berlin.de

its impacts[19,20]. For example, Burkina Faso's population is projected to face severe heat-related consequences in a future 2 °C global warming scenario[21,22]. Moreover, heat-related mortality is expected to increase by a factor of five by 2080[23]. Second, we differentiate climate change-induced heat stress increases and corresponding labour productivity loss among unskilled, semi-skilled, and skilled workers in different economic sectors across agroecological zones (AEZs). Heterogeneous climatic conditions make Burkina Faso suitable for analysing within-country disparities in heat stress-induced labour loss. Three agroecological zones can be distinguished in Burkina Faso: the Sudanian, Sudano-Sahelian, and Sahelian zones. The Sudanian zone is sub-humid with rainfall from 900 to 1200 mm/year, the Sudano-Sahelian zone is semi-arid with rainfall between 600 and 900 mm/year, and the Sahelian zone is arid and hottest with rainfall less than 600 mm/year[24]. Third, in contrast to previous studies, we distinguish the effects of heat stress-induced labour productivity loss on agriculture and household welfare across the three agroecological zones. In our welfare calculation, we further differentiate household groups based on their economic status (poor and non-poor) and residence (rural and urban). Lastly, our implementation of labour productivity shocks in the CGE model relies on a newly developed 2019 economy-wide database, called a social accounting matrix (SAM), which serves as a reference for Burkina Faso's economy.

## Results

This section first provides an overview of labour employment and income distribution across sectors in the reference scenario (base situation). Second, it presents the estimated heat stress-induced labour productivity shocks

implemented in the CGE model. Third, the effects of heat stress-induced labour productivity loss on factor prices, domestic production, household welfare, and macroeconomic indicators under a +1.5 °C scenario of meeting the Paris Agreement goal to keep warming below 2 °C and a +3.5 °C warming scenario are presented.

### Overview of labour income and employment based on the developed SAM

In 2019, unskilled and semi-skilled workers received 53% and 8% of the total revenue paid to labour, respectively (see Table 1). While representing only 5% (physical units) of total employment[25], skilled workers earned as much as 39% of the total labour revenue. Unskilled workers dominate all sectors in terms of physical labour employment. Regarding the labour income share, they dominate all sectors except the service sector. In the services sector, the income and employment shares of skilled workers are higher than in other sectors.

### Estimated heat stress-induced labour productivity loss

Table 2 presents the estimated heat stress-induced labour productivity shocks introduced into the CGE model. Labour productivity shocks in agriculture are differentiated across agroecological zones. Agriculture encompasses crops, livestock, fishing, aquaculture, and forestry. Accordingly, agricultural labour productivity shocks apply to all these subsectors. Labour productivity shocks to other sectors apply at the national level.

Regardless of the warming scenario, workers in the service sector who engage in light-intensity work are the least affected by heat stress (see Table 2 and Supplementary Fig. 1). This is followed by the manufacturing sector, where workers engage in moderate-intensity work (see Table 2 and Supplementary Fig. 1). The sectors most affected are agriculture, mining, and construction, which require heavy work. In agriculture, labour productivity loss is most substantial in the Sahelian zone, followed by the Sudano-Sahelian and Sudanian zones.

The productivity of skilled workers is least affected across sectors, as they often work indoors and are protected from solar radiation during their work time. Each additional temperature increase due to global warming further reduces the productivity of all workers, as indicated by the estimates for the +3.5 °C warming scenario.

### Factor price effects

Under the two warming scenarios, heat stress-induced labour productivity loss results in an economic downturn and reduces all factor prices except skilled labour, which is almost not affected, under the +1.5 °C warming scenario (Fig. 1). The less negative effect on wages for skilled workers in both scenarios is driven by relatively higher demand for skilled labour as this category is least affected by heat stress and thus gains in relative productivity compared to other labour categories. In the more drastic +3.5 °C warming scenario, skilled workers are also negatively affected due to their stronger productivity loss relative to the +1.5 °C scenario (Table 2). Compared to labour, capital and land become less scarce, resulting in their prices declining more.

## Table 1 | The distribution of labour income and employment (%) across skill levels per sector

| | Unskilled | Semi-skilled | Skilled | Total |
|---|---|---|---|---|
| Crops | 97.5 (98.6) | 1.0 (0.9) | 1.6 (0.5) | 100 (100) |
| Livestock | 96.7 (98.6) | 1.3 (0.9) | 2.1 (0.5) | 100 (100) |
| Forestry | 96.6 (98.6) | 1.3 (0.9) | 2.1 (0.5) | 100 (100) |
| Fishing and aquaculture | 96.8 (98.6) | 1.2 (0.9) | 2.0 (0.5) | 100 (100) |
| Mining | 98.2 (97.6) | 1.5 (2.0) | 0.3 (0.4) | 100 (100) |
| Food manufacturing | 98.2 (97.8) | 1.2 (2.2) | 0.6 (1.1) | 100 (100) |
| Non-food manufacturing | 98.6 (97.5) | 0.9 (1.7) | 0.5 (0.9) | 100 (100) |
| Construction | 88.5 (88.8) | 5.6 (5.5) | 5.9 (5.7) | 100 (100) |
| Services | 13.9 (81.7) | 13.0 (9.3) | 73.0 (9.0) | 100 (100) |
| Total | 53.4 (89.9) | 7.6 (5.3) | 39.0 (4.8) | 100 (100) |

For each sector, there are two figures per skill level. The first quantifies the distribution of labour revenue, and the second (in brackets) the distribution of labour employment across skill levels.

## Table 2 | Heat stress-induced labour productivity loss introduced as a shock in the model (in %)

| Activities | +1.5 °C warming | | | +3.5 °C warming | | |
|---|---|---|---|---|---|---|
| | Unskilled | Semi-skilled | Skilled | Unskilled | Semi-skilled | Skilled |
| Agriculture (Sudanian) | −9.9 | −9.9 | −4.0 | −26.6 | −26.6 | −16.7 |
| Agriculture (Sudano-Sahelian) | −10.4 | −10.4 | −5.5 | −26.9 | −26.9 | −19.6 |
| Agriculture (Sahelian) | −10.8 | −10.8 | −6.1 | −27.3 | −27.3 | −21.1 |
| Mining | −10.4 | −10.4 | −5.4 | −26.9 | −26.9 | −19.6 |
| Manufacturing | −9.3 | −9.3 | −2.4 | −25.6 | −25.6 | −13.5 |
| Construction | −11.0 | −11.0 | −8.7 | −26.9 | −26.9 | −24.3 |
| Services | −7.1 | −7.1 | −0.3 | − 21.5 | −21.5 | −5.3 |

The table summarizes the computed labour capacity loss estimates, which were introduced as labour efficiency shocks in the simulation model.

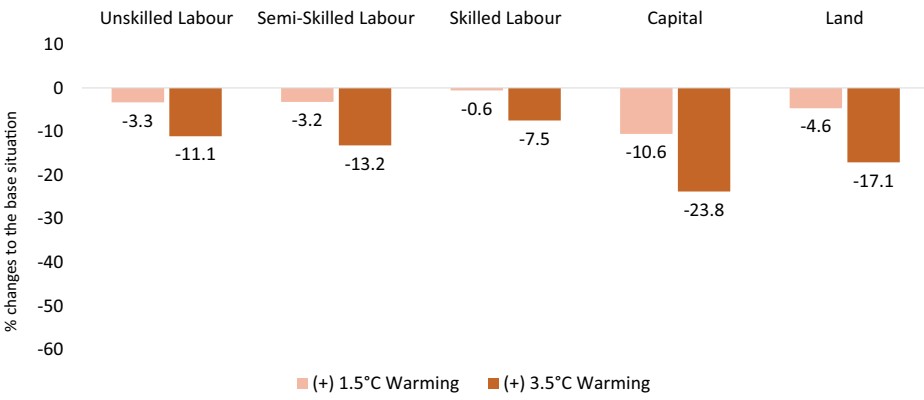

**Fig. 1 | The effects of heat stress-induced labour productivity loss on factor prices.** Heat stress simulation results for factor price relative to the reference situation, expressed as a percentage, under +1.5 °C (light orange) and 3.5 °C (dark orange) warming scenarios.

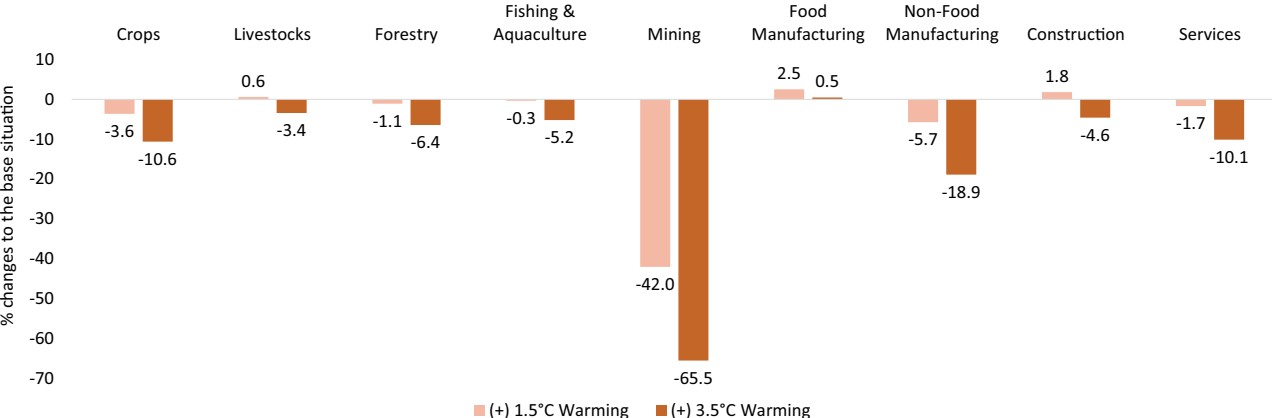

**Fig. 2 | The effects of heat stress-induced labour productivity loss on domestic production.** Heat stress simulation results for domestic production, relative to the reference situation, expressed as a percentage, under +1.5 °C (light orange) and 3.5 °C (dark orange) warming scenarios.

## Production effects

The production effects of heat stress-induced labour productivity loss differ among sectors, depending on whether the sectors are labour-intensive or not (Fig. 2). Mining, non-food manufacturing and crop production are most adversely affected in both warming scenarios. This is due to these sectors' considerable dependency on unskilled workers, whose productivity declines the most. The service sector shrinks the least, as it relies primarily on indoor skilled workers who are substantially less affected by heat stress than others.

The capital-intensive sectors are less negatively affected or even grow. For example, construction and food manufacturing grow under +1.5 °C warming due to the declined capital costs compensating for the lower labour productivity and effects of the overall downturn of the economy. In the +3.5 °C warming scenario, the negative effects of heat stress outweigh the lower capital costs, leading to a decline in construction outputs. Similarly, food manufacturing only maintains its initial production level. Other capital-intensive sectors, such as livestock and fishing/aquaculture, also benefit from lower capital costs, resulting in smaller production losses than for crops. Forestry, a land-intensive sector, declines considerably in both warming scenarios due to the reduced demand for forestry products as intermediate inputs in other sectors (Supplementary Fig. 2).

Analysis of the crop sector shows that overall production decline is primarily driven by cotton (Fig. 3). Cultivated exclusively in the Sudanian and Sudano-Sahelian zones, the cotton (The cotton sector comprises both raw cotton production and ginning.) sector declines by an average of 7% and 16% under two warming scenarios, respectively. Since cotton constitutes over 30% of total crop output, its decline substantially impacts aggregate production.

Maize, rice, and tubers are also exclusively produced in the Sudanian and Sudano-Sahelian zones. The higher the heat stress level, the greater the adverse effects on these crops. The production of sorghum, millet, fonio, peanuts, legumes, fruits, and vegetables shows a similar pattern, being more affected in the Sudano-Sahelian than in the Sudanian zone. Some of these crops, including rice, peanut and sorghum, even grow in the Sudanian zone. Compared to the Sudano-Sahelian zone, these crops are less labour-intensive in the Sudanian zone, resulting in lower or negligible heat stress effects. This is also the case for sorghum and peanuts in the Sahelian zone, which are either unaffected or increasing in both scenarios.

The aggregate crop production decreases most in the Sudanian and Sudano-Sahelian zones under both warming scenarios, although workers in these zones experience less productivity loss than those in the Sahelian zone (Fig. 4). Aggregate crop production in the Sudanian and Sudano-Sahelian zones declining most is due to a composition effect, as the share of the most affected crop, cotton, is highest in these regions. These two zones account for more than 80% of total crop production in the base situation. Highly affected crops, which are dominant in these zones, result in larger labour productivity shock effects at the aggregate level, despite smaller labour productivity shocks than in the Sahelian zone. In the livestock sector, production declines more in the Sahelian zone compared to other zones, reflecting the higher labour productivity shocks induced by heat stress in that zone (see Supplementary Fig. 3).

## Household welfare effects

Labour productivity loss induced by heat stress affects household groups differently (Fig. 5). Generally, the heterogeneity of welfare effects stems from differences in the composition of factor income and consumption, as well as differences in heat stress-induced labour productivity losses across agroecological zones (Supplementary Fig. 1).

Rural poor households lose welfare under both warming scenarios. This result is primarily due to reduced income from supplying land, unskilled labour, and semi-skilled labour (Supplementary Fig. 4). The

**Fig. 3 | The effects of heat stress-induced labour productivity loss on crops across AEZs.** Heat stress simulation results for each crop, relative to the reference situation, expressed as a percentage, under +1.5 °C (light orange) and 3.5 °C (dark orange) warming scenarios. This presents percentage change results from top to bottom for maize, rice, sorghum, millet, and fonio, as well as tubers, cotton, peanuts, legumes, fruits, and vegetables, respectively.

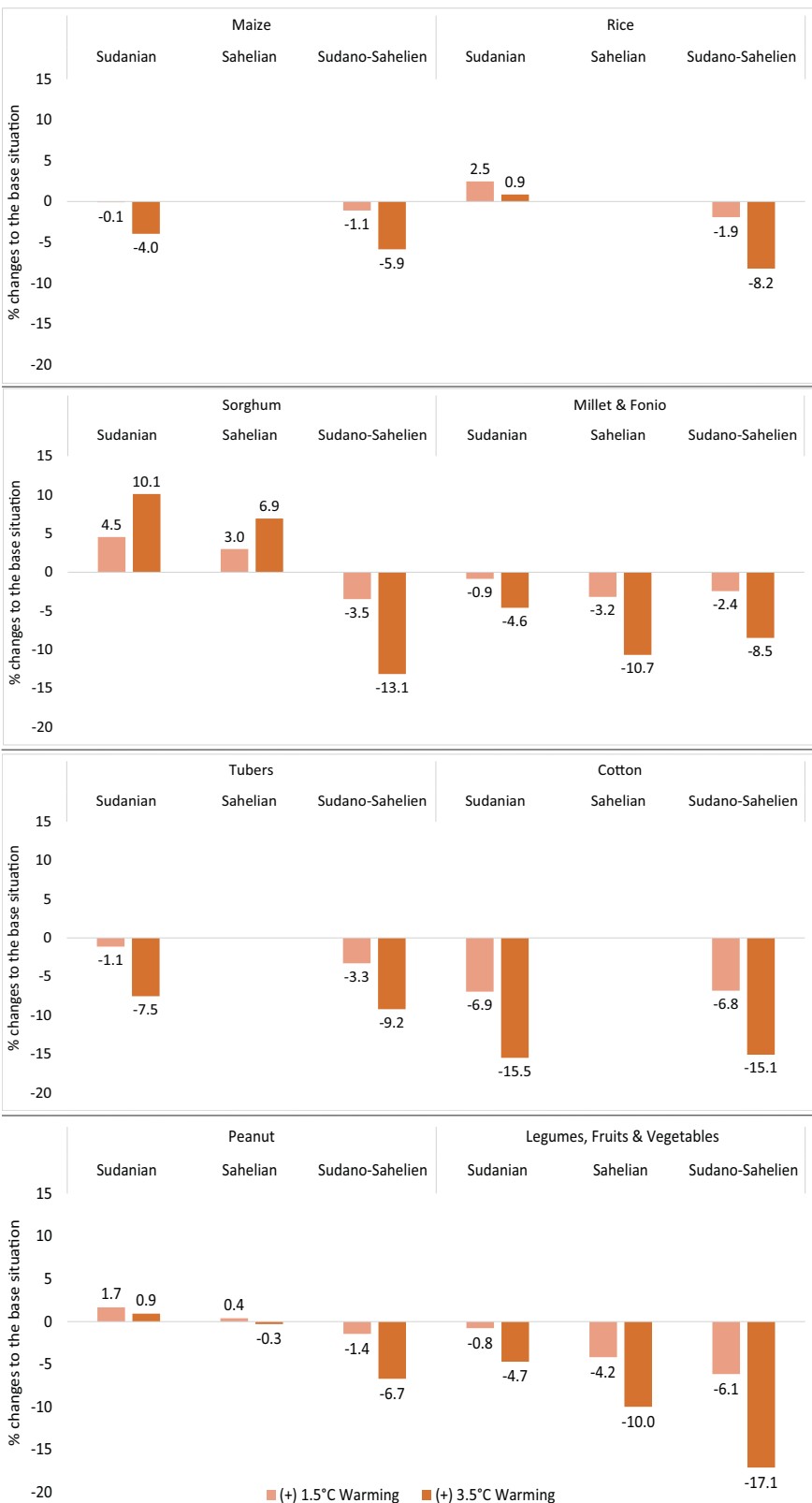

greater welfare loss among these households is due to their heavy dependence on these production factors, in contrast to capital and skilled labour. The welfare of urban poor households is similarly affected, but to a lesser extent than that of their rural poor counterparts. Their welfare loss is also driven by reduced income, mainly due to declining wages for unskilled and semi-skilled labour. Besides the income effect, both rural and urban poor

households face higher prices for crop commodities, which dominate their expenditure pattern.

Both rural and urban non-poor households are unaffected or experience welfare gains in the +1.5 °C warming scenario, primarily driven by declining consumer prices (Supplementary Fig. 5) for commodities such as livestock, fishing/aquaculture, and services, which collectively represent a substantial share of their consumption expenditures (Supplementary

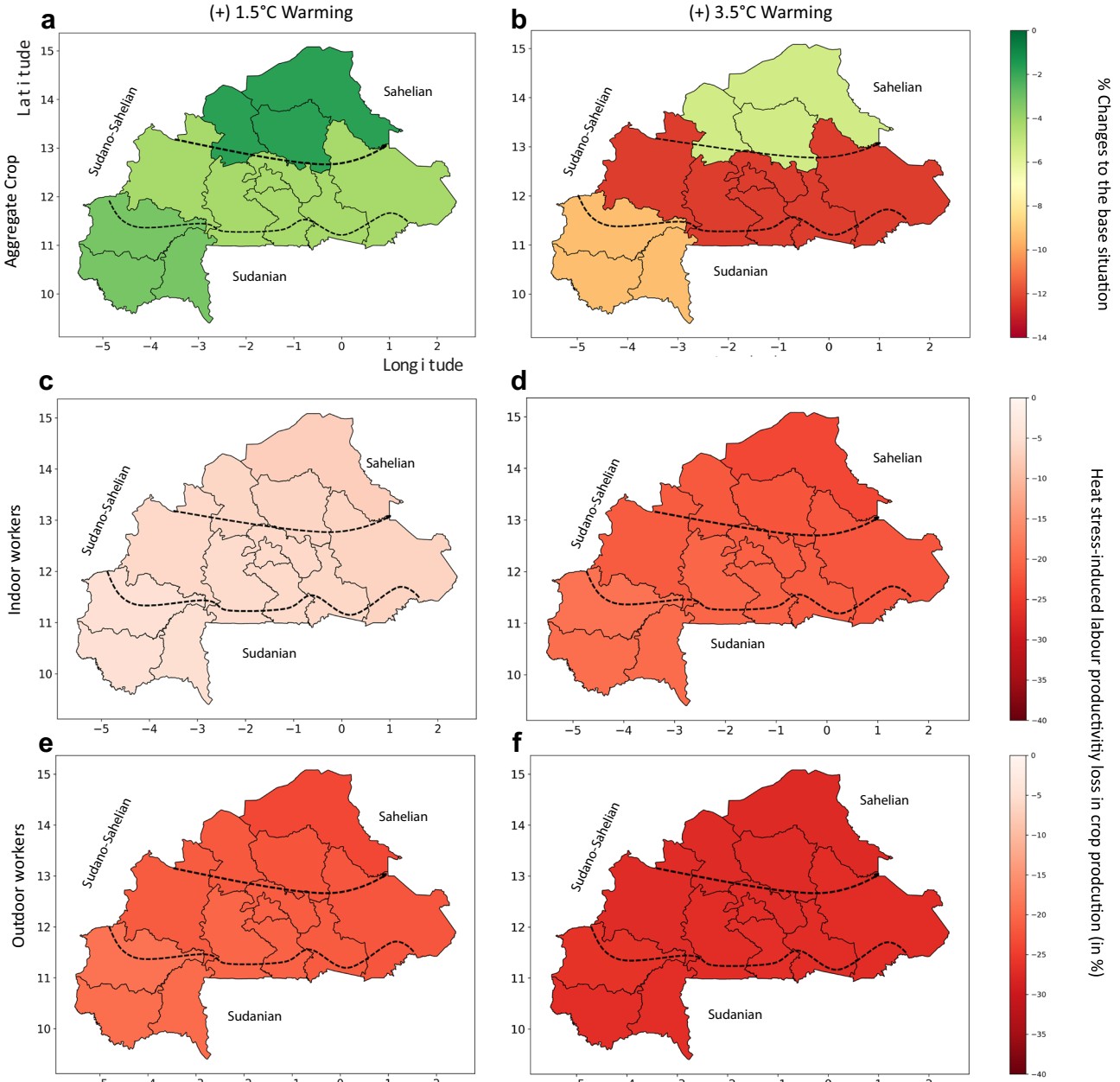

**Fig. 4 | The effects of heat stress-induced labour productivity loss on crop production across AEZs.** Heat stress simulation results for aggregate crop across agroecological zones, relative to the reference situation, expressed as a percentage, under +1.5 and 3.5 °C warming scenarios. The top two maps, **a**, **b** show aggregate crop changes across agroecological zones in Burkina Faso under +1.5 and 3.5 °C warming scenarios, respectively. The dark green indicates a smaller heat stress effect, while the dark red shows a stronger heat stress effect on the aggregate crop. The four maps at the bottom show the corresponding labour capacity loss across agroecological zones, with **c**, **d** illustrating the capacity loss of indoor workers under +1.5 and 3.5 °C warming scenarios, respectively. **e**, **f** Illustrate the capacity loss for outdoor workers under the two warming scenarios, respectively. Lighter red indicates smaller labour capacity loss, while darker red indicates the opposite.

Table 1). Compared to poor counterparts, rural and urban non-poor households have a lower expenditure share for food commodities (Supplementary Table 1) and are therefore less affected by increasing crop commodity prices (as shown in Supplementary Figs. 5 and 6).

Household groups are affected differently across agroecological zones (Fig. 5). Rural and urban poor households in the Sahelian zone lose slightly more welfare than their counterparts in other agroecological zones across both warming scenarios. This is due to their factor income from unskilled and semi-skilled labour being relatively higher compared to their counterparts in other zones. Among the non-poor households, the Sahelian zone is the most affected, because both rural and urban non-poor hold a relatively higher income share from unskilled compared to skilled labour than their counterparts in

other zones. The wages of unskilled and semi-skilled labour are more affected than those of skilled labour, which is reflected in the welfare changes of these households in the Sahelian zone.

**Macroeconomic effects**

Figure 6 presents the macroeconomic effects caused by heat stress-induced labour productivity loss. The economy shrinks due to declining domestic production, which adversely affects GDP, resulting in drops of about 9% and 20% in the +1.5 and 3.5 °C warming scenarios, respectively. The GDP loss can primarily be attributed to the substantial decline in crop production (especially cotton), mining, and non-food manufacturing, which collectively contribute a substantial share of the overall GDP. Exports drop considerably due to lower domestic production, especially of key export commodities

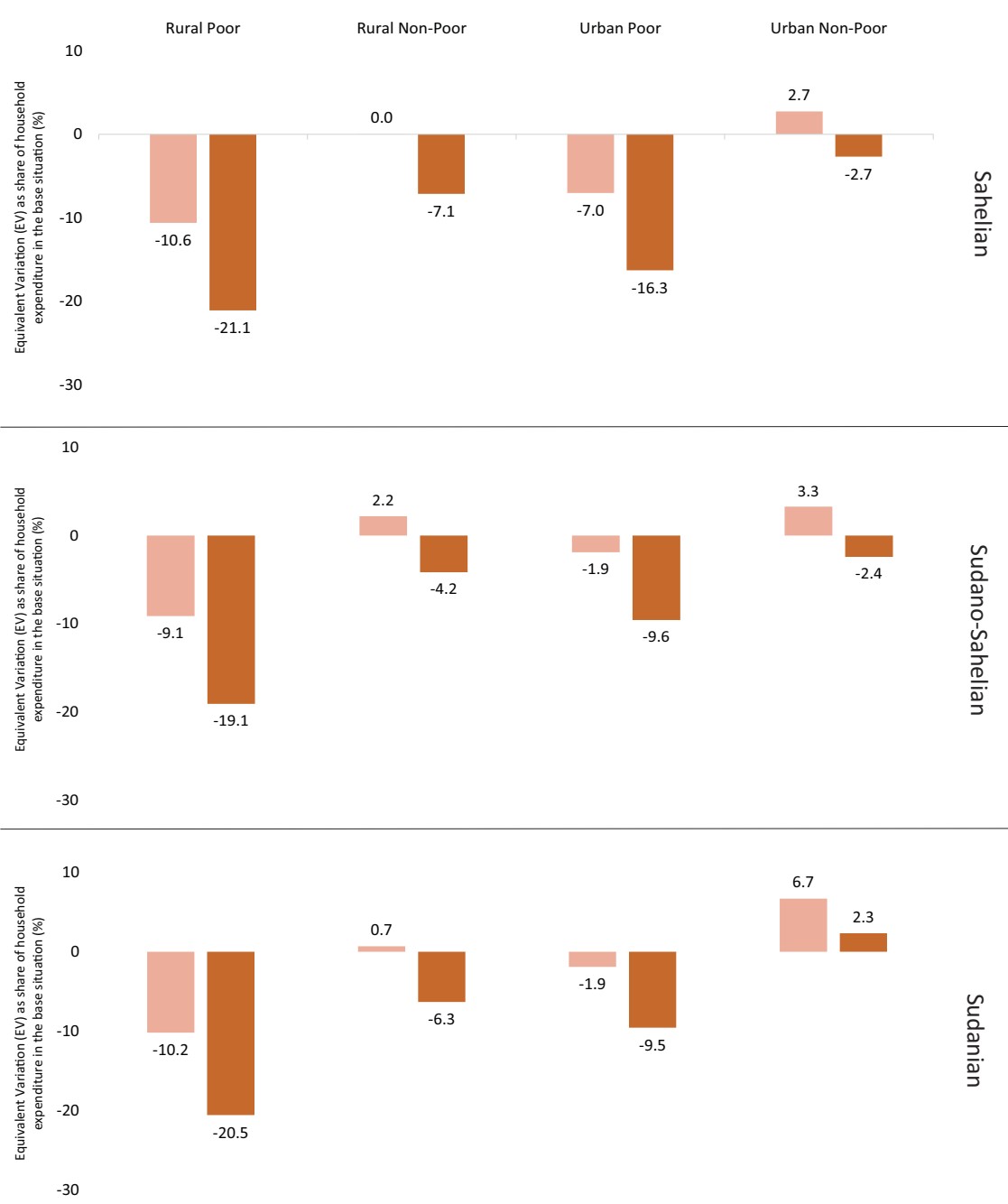

**Fig. 5 | The effects of heat stress-induced labour productivity loss on household welfare across AEZs.** Heat stress simulation results for household welfare, measured using Equivalent Variation (EV), under +1.5 °C (light orange) and 3.5 °C (dark orange) warming scenarios. Equivalent variation (EV) is the measure used for household welfare. We expressed the EV as a percentage of household income in the base situation. As shown in the figure, we report household welfare by location and economic status across agroecological zones, starting with the Sahelian at the top, the Sudano-Sahelian in the middle, and the Sudanian at the bottom.

such as cotton (Fig. 3) and mining (Fig. 2). In both warming scenarios, fewer commodities are imported, largely due to lower demand for intermediate inputs. Under +1.5 °C warming, overall household consumption increases slightly due to the slight welfare gain for non-poor households. Investment declines under both scenarios due to the overall economic downturn.

We report the results of various sensitivity analyses for the macro-economic changes under the +1.5 °C warming scenario to demonstrate the robustness of our model results (see Fig. 7).

The result patterns remain the same when the substitution possibility (substitution elasticity (SE) = 1.5) between labour types decreases or increases by 50%. However, the magnitude of the results slightly differs (Fig. 7). For example, domestic production effects are stronger when the elasticity of substitution is lower. Conversely, they are weaker when the elasticity of substitution is higher. This is due to the increased flexibility of sectors to substitute between less affected skilled labour and more affected unskilled and semi-skilled labour types, as well as capital, thereby lowering the negative effects of heat stress on production. The results also deviate slightly, with higher labour-capital substitutability (increased by 50%). The effects of labour heat stress are lower, as less productive labour can be better compensated for due to the higher labour-capital substitutability (see Fig. 7).

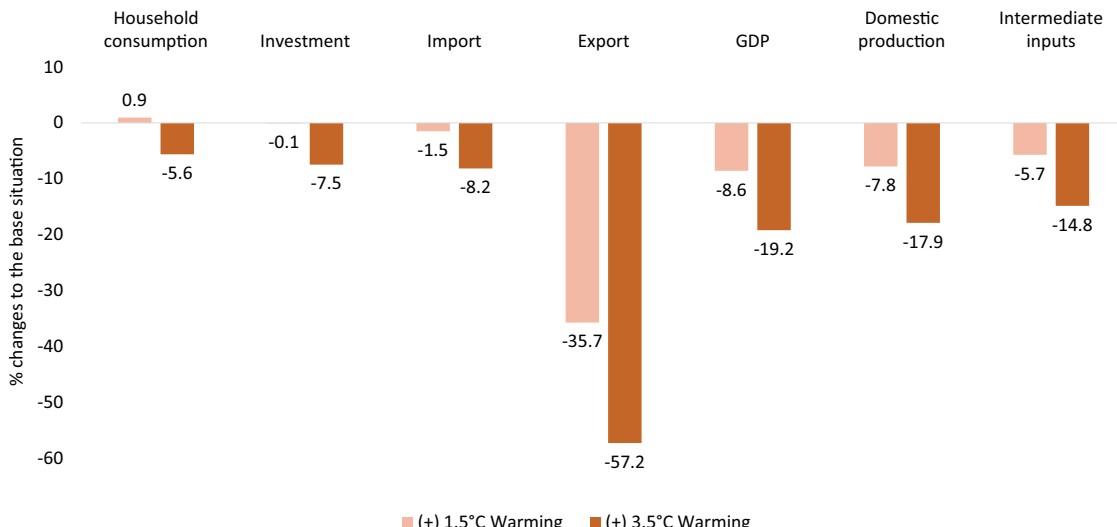

**Fig. 6 | The effects of heat stress-induced labour productivity loss on the economy.** Heat stress simulation results for macroeconomic indicators relative to the reference situation, expressed as a percentage, under +1.5 °C (light orange) and 3.5 °C (dark orange) warming scenarios.

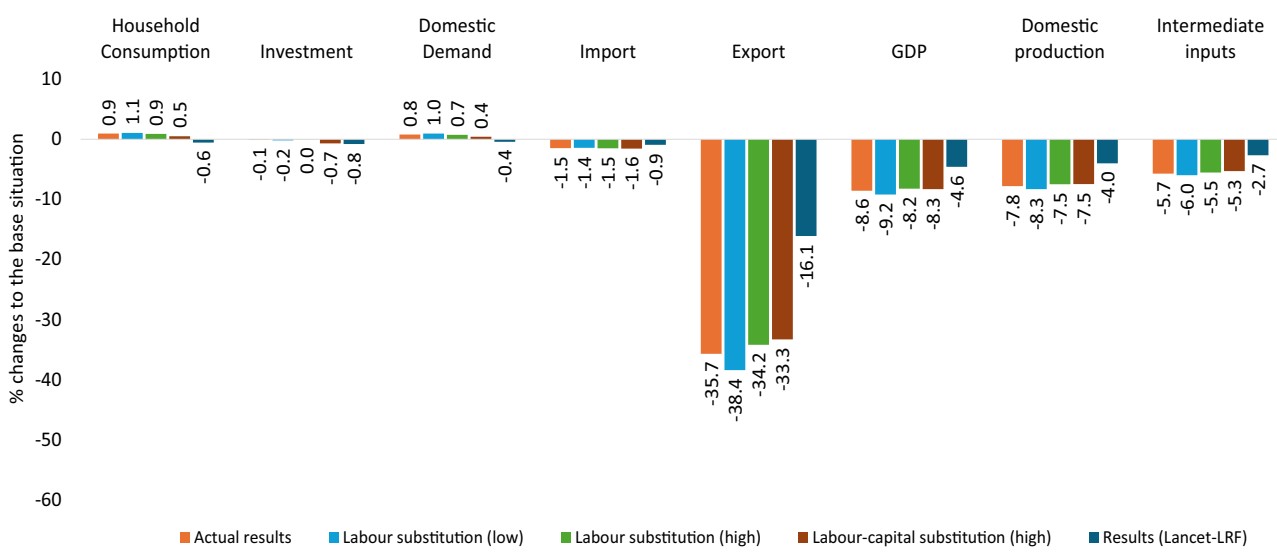

**Fig. 7 | Sensitivity analysis results for macroeconomic effects under +1.5 °C warming scenario.** Heat stress simulation results for macroeconomic indicators relative to the reference situation, expressed as a percentage, based on different sensitivity analyses, under a +1.5 °C warming scenario. The first bar, in orange, represents the macroeconomic results of the model. Light blue, light green, brown, and dark blue represent macroeconomic results for lower labour substitution, higher labour substitution, higher labour-capital substitution, and the Lancet labour response function sensitivity, respectively.

We also repeat the whole analysis using an alternative labour response function as derived from the "Hothap" project[26] and modified and adopted in the Lancet Countdown report[27]. The results remain qualitatively the same when using the ISO labour response function. Here, we present the results for the economy-wide effect, noting that similar findings apply to other types of impact. The labour shock effects are lower when using the Lancet labour-response function (Lancet-LRF) (Fig. 7). This is because the ISO function is designed to protect the health and safety of workers worldwide. In contrast, the Lancet function is derived from real-world field studies on highly heat-acclimated labour in subtropical countries, where workers may tend to work beyond their safety capacity (see the "Methods" section). As a result, the Lancet function estimates lower productivity loss compared to the ISO function (Supplementary Table 2). However, both functions generate similar patterns of macroeconomic effects across different aspects of the economy, suggesting that our results are robust to the choice of labour response functions.

## Discussion

Our study analyses the heterogeneous effects of human heat stress across agroecological zones within a country. Accounting for this heterogeneity in the context of Burkina Faso, the results show that heat stress impacting labour productivity affects crop production and household welfare differently across agroecological zones. Heat stress-induced labour productivity loss has the strongest negative effects on the aggregate crop sector in the Sudanian and Sudano-Sahelian zones, even though crop labour productivity loss is less in these zones than in the Sahelian zone (see Fig. 4). This is due to the higher share of cotton, the most affected crop, in overall crop production in these two regions. This highlights an essential composition effect to be aware of when appreciating heat stress effects on aggregate sectors. The Sudanian and Sudano-Sahelian zones are the major crop-producing regions, accounting for over 80% of total domestic crop production. Contrary to what could be expected, these results show that these two zones, although not as strongly affected by heat stress as the Sahelian zone, are of special interest for targeted heat stress mitigation strategies to protect the

crop sector and continue supplying foods to private households, and generating foreign income through cotton exports.

We conduct a disaggregated household welfare analysis, which has not been part of previous studies[10–13]. Rural poor households are most negatively affected in Burkina Faso, given their involvement in outdoor economic activities, which shrink due to heat stress. The non-poor households could maintain their livelihoods under a 1.5 °C warming, but welfare declines with a more severe +3.5 °C warming. Being less affected compared to poor households is attributable to their higher income share of skilled labour and to their consumption patterns, with relatively high shares of products declining or increasing less in prices than crop commodities (see Supplementary Fig. 5). These comprise commodities such as livestock, fishing/aquaculture and services, accounting for a substantial share of their expenditures (see Supplementary Table 1). In contrast, they allocate a very small share of their consumption to crop commodities (see Supplementary Table 1), which makes them less negatively affected by increased crop prices (see Supplementary Fig. 5). Considering agroecological zones, our results indicate that households, whether rural or urban, poor or non-poor lose more welfare in the Sahelian zone than in other zones (see Fig. 5 and Supplementary Fig. 7). This outcome could explain the recorded climate change-induced migration, showing farmers' movement from the hotter Sahelian zone to the other zones as a heat adaptation strategy[28].

We infer three key conclusions. First, considering regional differences across agroecological zones is relevant due to heterogeneous labour productivity shocks and household income and expenditure patterns, as well as relevant composition effects according to regional agricultural specialization. Second, heat stress-induced labour productivity loss exacerbates income inequality, affecting the welfare of poor households more than that of non-poor households. Third, to withstand the effects of climate change-induced heat stress, there is a need for economic diversification, shifting from vulnerable sectors such as cotton to more diverse and resilient production structures. In the case of Burkina Faso, the economy is highly labour-intensive and relies mainly on cotton and mining to generate most of the foreign revenues.

Our study provides insights into the potential sectoral, economy-wide, and household welfare effects of global warming through heat stress, but it has some limitations:

- Like all heat stress metrics, WBGT faces various limitations[29,30], which may lead to under- or overestimating labour productivity loss from heat stress. Still, it has been found to perform as well as, or better than, other metrics in evaluating the physiological impact of heat exposure[31,32].
- We do not explicitly cover the time dimension of heat stress evolution under climate change or economic development.
- In our analysis, we have excluded heat stress effects experienced by urban households at night. Urban heat islands pose challenges to urban households[33,34]. Besides, they incentivize the continuous use of cooling devices, requiring additional energy costs for urban households[35]. Accounting for these aspects could result in a more negative welfare effect for urban households compared to what we obtained in this study.
- We do not account for the direct effects of heat stress on plants[36,37] and animals[38–41]. Assessing the combined effect of heat stress on plants, animals, and humans would likely find more severe economic effects. In both scenarios, we found declining prices for livestock and fishing commodities. Direct heat stress effects on animals, which are not modelled in this study, could result in lower production and higher prices for these commodities than presented here, as well as more negative economy-wide effects.
- In our model, productive activities compete for factors mobile across sectors. And indeed, labour could move between regions due to the increased temperature[42,43]. To depict explicitly the regional movement of labour incentivized by heat stress, different labour types would need to be disaggregated by climatic zones in the developed economy-wide database. Furthermore, a model extension would be required to

capture heat stress-induced labour movement across climatic zones, for example, by including factor mobility functions[44].

- We have grouped Burkina Faso's regions into agroecological zones for agricultural production and household group disaggregation. This grouping is not fully accurate, as a small portion of some regions overlaps with other agroecological zones (Supplementary Fig. 8). Sub-region-level data (not available during this research) could be useful to distinguish agroecological zones more precisely.
- Single-country CGE models, including the one applied in this study, typically operate under the assumption of fixed international prices. This implies that our analysis only accounts for the effects of labour heat stress within Burkina Faso, but not for world market price-mediated effects of heat stress in other countries. However, global warming is expected to affect labour productivity in many countries. This may shift international prices, especially in heat stress-sensitive sectors, such as, for example, cotton and mining, adding to the effects of changes in domestic labour productivity. For example, a rise in world market prices, e.g., for cotton and mining commodities, could incentivize increased production in Burkina Faso. This response could partially offset the production declines observed under the assumption of fixed international prices, especially in export-intensive sectors. Higher global prices may also result in higher prices of imported intermediate goods, such as fertilizers and machinery. Sectors dependent on these inputs may therefore decline, potentially amplifying the overall negative impact on GDP. Analysing such a scenario would require estimates of international price changes, ideally derived from a global CGE model under the same warming scenarios as in this study, which is currently not available.

The results highlight the most vulnerable sectors, household groups, and agroecological zones that require specific consideration in the development and implementation of heat stress adaptation and mitigation policies in Burkina Faso. We identify the following policy implications:

- The strong adverse effects on labour-intensive sectors, mostly cotton and mining, signal the need for diversification into less labour-intensive production systems. For example, investments in machinery could be made in the cotton sector, rendering it more mechanized by increasing the use of tractors and harvesters. More automation in the mining sector could help offset the negative effects of climate change-induced declines in labour productivity. As shown by the sensitivity analysis above, a shift from labour to more capital-intensive production could mitigate the overall negative effects of labour heat stress.
- A continuous implementation of the ongoing "Great Green Wall"[45] in Burkina Faso could be beneficial in creating a cooler environment, specifically in the Sahelian zone, and limiting the effects of climate change-induced heat exposure on workers.
- Promoting agroforestry initiatives mostly in the crop-producing Sudanian and Sudano-Sahelian zones through tree planting on farms could limit the direct heat effects on farming workers, supporting sustained food production and export income[46].
- Burkina Faso's National Climate Change Adaptation Programme (NCCAP) emphasizes the need for enhancing Early Warning Systems to mitigate heat-related effects of climate change[47]. These were already effective in response to the recent heat waves recorded in the country[48].
- The country would benefit from coping measures such as the promotion of cool roofs[49–51], green roofs[52], and sustainable adoption of cooling devices such as fans and air conditioning units[35] in workplace premises and household units.

These measures would require a strong government commitment towards their implementation. Beyond Burkina Faso, the policy implications are partially transferable to other Sahelian West African countries such as Mali and Niger sharing almost similar climatic and economic conditions as Burkina Faso.

## Methods

The methodological approach comprises four steps. First, we developed an economy-wide database for Burkina Faso, which serves as the benchmark for calibrating the single-country CGE model employed. Second, we estimated heat stress using the wet bulb globe temperature (WBGT) based on projected climatic data on temperature, wind speed, humidity, and solar radiation under future global warming scenarios. Third, we employed a labour-response function to estimate climate change-induced labour productivity loss using the estimated WBGT. Lastly, we implemented the estimated heat stress-induced labour productivity loss through the labour efficiency parameter in the CGE model to analyse economy-wide implications.

### Data

We developed an economy-wide database, known as a Social Accounting Matrix (SAM), for Burkina Faso for the year 2019. This year is explicitly selected because it predates the impacts of COVID-19 and the rise in terrorist activities in recent years. This SAM is built using data from the 2013 SAM for Burkina Faso[53], the unpublished 2019 SAM for Burkina Faso, developed by the National Bureau of Statistics (INSD in French), the published 2017 SAM for Burkina Faso[54], Burkina Faso National Statistics[25,55,56], the Organisation for Economic Co-operation and Development (OECD)[57], World Bank Group[58], and the Food and Agriculture Organisation (FAO)[59].

The SAM records transactions among economic agents and sectors into 89 accounts, with 47 sectors producing 18 commodities. The sectors comprise 14 agricultural activities in three agroecological zones, producing 14 commodities, as well as food and non-food manufacturing, which produces one manufacturing commodity. Additionally, mining, construction and services each produce one commodity. In our SAM, the Sahelian zone comprises (a) Sahel, (b) Nord and (c) Centre-Nord. The Sudano-Sahelian comprises (a) Boucle de Mouhoun, (b) Centre, (c) Est, (d) Centre-Est, (e) Centre-Sud, (f) Centre-Ouest, and (g) Plateau-Central. And the Sudanian comprises (a) Haut-Bassins, (b) Cascades, and (c) Sud-Ouest (Supplementary Fig. 8).

Sectors employ five factors: land, capital, and three labour categories. The three labour types are distinguished by their education level. Unskilled workers have no formal or primary education and dominate the economy, with 90% of the employment. The semi-skilled workers have post-primary (lower secondary) education, while skilled workers have upper secondary education and above, representing 5.2% and 4.8% of the workforce, respectively[25]. This disaggregation enables the differentiation of heat stress effects on each type of labour.

This SAM introduces land as a production factor used by agricultural activities. The use of farmland was not depicted as a separate account in the existing SAMs for Burkina Faso, but was considered as part of the overall capital used by activities. The farmland tenure system in Burkina Faso makes it challenging to reflect land valuation accurately in the SAM. Inheritance is the most common form of agricultural land tenure, followed by land transfers through donations and marriage, while farmland purchases account for <1% of total farmland[56]. We adopted an opportunity cost approach to incorporate the land account in this SAM. We applied the 2019 annual interest rate of 6.1% from the World Bank Indicator data[58] to an average land purchase price of 500,000 F.CFA per hectare, multiplied by the total area (in hectares) of crops and forests in 2019[59]. The land purchase price was determined based on expert opinions.

According to location, poor and non-poor, and agroecological zones, the SAM distinguishes 12 household groups using data from the 2018 Household Survey Data[56]. These groups comprise rural-poor, urban-poor, rural-non-poor, and urban-non-poor in the three agroecological zones (Supplementary Fig. 8). Additionally, the SAM accounts for government, enterprises, savings/investments, and the rest of the world, as well as indirect taxes (related to production and sales) and direct taxes (on income).

### Model

The study employs the STAGE version 2 model, developed by McDonald and Thierfelder[60], which is calibrated to the developed 2019 SAM for Burkina Faso. STAGE is a comparative-static single-country CGE model. It offers the possibility to simulate, on a national scale, the future equilibrium state of an economy after a shock. In this context, a shock can refer to an economic crisis, a new policy, or heat stress-induced labour productivity loss, as observed in our study. In STAGE, households maximize utility by demanding bundles of commodities, using the Stone–Geary utility function. This function accounts for subsistence consumption, making it well-suited to representing consumption behaviour in a developing country like Burkina Faso, where many consumers live in poverty. In addition, domestic production is modelled as a three-stage production (Supplementary Fig. 9). In the first stage, aggregate intermediate and aggregate value added (primary inputs) are combined using constant elasticity of substitution (CES) or Leontief technology. In the second stage, aggregate intermediate inputs are combined using Leontief technology. This implies that activities demand intermediate inputs in fixed proportions. Aggregate primary inputs such as land, capital, and labour are combined into aggregate value added using CES technology, allowing for substitution between production factors. In the third stage, natural primary inputs are combined using CES technology to produce aggregate primary inputs. Thus, activities employ aggregate labour, through a CES combination of unskilled, semi-skilled, and skilled labour, with different labour skill levels being substitutes to some degree. Additionally, the model allows for multiple activities to produce a single commodity, as seen in the case of agriculture and manufacturing in this application.

The STAGE model operates under four closure conditions representing the fundamental characteristics of Burkina Faso's economy. These conditions include the closure of the foreign exchange, investment-savings, government, and factor markets.

**Foreign exchange market closure.** As a member of the West African Economic and Monetary Union (WAEMU), Burkina Faso uses the West African CFA, a currency pegged to the Euro (1 Euro = 655.957 F.CFA). Accordingly, we fixed the exchange rate in the model, and the external balance (foreign savings) is flexible.

**Investment-saving closure.** We fixed the initial share of investments in final demand such as not to affect future development through changes in current period investments. In other words, households, enterprises, and foreign savings are flexible to equate the total investment in the economy.

**Government closure.** Government savings (in absolute terms) and expenditure (share of final domestic demand) are fixed. Household income tax rates adjust endogenously in the model to keep the government savings and expenditure share at its initial.

**Factor market closure.** The initial total factor supply in the economy is fixed and fully absorbed by sectors through changes in the wage rates. Furthermore, factors are assumed to be mobile across productive activities. Formally non-employed workers in Burkina Faso still contribute to the economy by working in the informal sector, which is the reason for the large employment. The model closures end with the consumer price index (CPI) defined as the model numeraire.

### Estimation of heat stress and labour productivity loss

Heat stress-induced labour productivity shocks are estimated following the same workflow as implemented in ref. 13. Here, we outline the procedure briefly. First, heat stress is quantified by wet-bulb globe temperature (WBGT), as calculated from the Liljegren et al.[61] approach using the Python implementation developed by Kong and Huber[62]. Indoor WBGT is calculated using the Bernard[63] approach, assuming no solar radiation and 1 m/s wind speed (Eq. (8) in ref. 64). This choice aligns with the recommendation

made by Lemke and Kjellstrom[64]. Second, indoor and outdoor labour capacities are estimated from WBGT using the ISO labour response function (Eqs. (1) and (2) in ref. 13) for light, moderate, and heavy intensity work, equivalent to metabolic rates of 200, 300, and 400 W. This calculation is initially performed at a 3-hourly scale, which is then aggregated to daily and annual average scales while only accounting for daytime hours. Next, we estimate the climatological annual average labour capacity at the grid cell level for the baseline period of 1961–1990 and for periods corresponding to global warming levels of 1.5 and 3.5 °C relative to the baseline. The labour capacity shock due to climate change-induced heat stress is then quantified by comparing the warmer periods against the baseline. Finally, we aggregate grid-cell-level labour capacity shock to the level of agroecological zones or the entire country for all combinations of indoor/outdoor environments and work intensity. We classify agriculture, construction, and mining as heavy work, manufacturing as moderate work, and services as light work, following the work intensity classification in the ISO7243 standard[65]. This generates agroecological zone-level labour shocks for each economic sector in Burkina Faso. The calculations described are conducted for each climate model and then aggregated to obtain the model ensemble mean, which is used as input for the CGE model.

## Implementation of heat stress-induced labour productivity loss in the model

We implemented heat stress-induced reduction in labour productivity through the labour efficiency parameter in the CGE model. Thus, we simulated labour productivity shocks that may occur in the future in Burkina Faso's economy, based on the current state of the economy (the reference year 2019), using calculated heat stress-induced labour productivity losses under the +1.5 and +3.5 °C warming scenarios. The +1.5 °C warming scenario assumes meeting the Paris Agreement goal of keeping the global temperature rise below 2 °C, while the +3.5 °C scenario assumes failure to meet this goal. We account for the daytime labour capacity loss estimates, assuming that, in Burkina Faso, labour is employed only during the day, irrespective of the sector. Moreover, we assume that unskilled and semi-skilled workers are mostly outdoors, while skilled workers are indoors. Therefore, we used the outdoor labour capacity loss estimates to stimulate the productivity loss of unskilled and semi-skilled workers in the two scenarios. The corresponding indoor estimates were used for skilled workers.

Regarding agriculture, we used the estimates of indoor and outdoor labour productivity shocks for each agroecological zone.

## Sensitivity analysis

In our model, productive activities can substitute between unskilled, semi-skilled, and skilled labour given an elasticity of substitution of $\sigma = 1.5$, as suggested by Ciccone and Peri[66]. We conducted two sensitivity analyses to determine to what extent the degree of labour substitution affects our results. In the first sensitivity analysis, we reduce the labour substitution elasticity by 50% (elasticity of substitution, $\sigma = 0.75$). In the second sensitivity analysis, we increase this elasticity by 50% (elasticity of substitution, $\sigma = 2.25$). We conducted a third sensitivity analysis, which increases labour-capital substitutability by 50%. Finally, we repeated the entire analysis using an alternative labour response function derived from the "Hothap" project, based on two empirical studies: one on gold mining workers in South Africa[67] and another on rice harvesters in India[68]. The "Hothap" function was slightly modified and adopted in the Lancet annual countdown report[27], and the Lancet modified version (Eq. (3) in ref. 13) is used in this work.

## Data availability

The database[69] used in this study is available at Zenodo: https://doi.org/10.5281/zenodo.15603366. Simulation model results and processed data used for generating graphs and figures are available at Zenodo: https://doi.org/10.5281/zenodo.16088699.

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

## Acknowledgements
Deutsche Forschungsgemeinschaft (DFG) funded the study as part of the Research Unit FOR 2936: Climate Change and Health in Sub-Saharan Africa. Additionally, the NIH grant R01HD104835, NASA FINESST Grant 80NSSC22K1544, and NSF Grant 2020635, "AccelNet:GLASSNET: Networking Global to Local Analyses to Inform Sustainable Investments in Land and Water Resources," funded the work on WBGT and labour capacity loss computation. Lastly, we acknowledge support by the Open Access Publication Fund of Humboldt-Universität zu Berlin.

## Author contributions
M. Houessou conceptualized the research, developed the simulation and carried out the simulations under the supervision of Z.E. and H.G. Q.K. computed WBGT and labour capacity loss under the supervision of M. Huber. M. Houessou drafted most of the manuscript, with contributions from Z. E., Q.K., H.G., and M. Huber.

## Funding

## Competing interests
The authors declare no competing interests.
