## [Transparent Peer Review file · Communications Earth & Environment]

Heat stress causes economic and welfare disparities across agroecological zones in Burkina Faso

Corresponding Author: Mr Martial Houessou

Version 0:

Decision Letter:

Dear Mr Houessou,

Your manuscript titled "Welfare Implications of Heat Stress-Induced Labour Productivity Loss under Global Warming Scenarios in Burkina Faso" has now been seen by 2 reviewers, whose comments are appended below. You will see that they find your work of some potential interest. However, they have raised quite substantial concerns that must be addressed. In light of these comments, we cannot accept the manuscript for publication, but would be interested in considering a revised version that fully addresses these serious concerns. Specifically, we ask you to:

- 1) Provide compelling new insights into the economic consequences of heat stress-induced labor productivity losses with global warming at national and subnational scales in Burkina Faso, and discuss the relevance of your findings for heat stress adaptation and mitigation policies
- 2) Discuss further spatial differences and drivers of sectoral compositions and vulnerabilities
- 2) Fully justify your methodology and approach, including choice of variables and model calibration, and discuss the limitations of your study

We hope you will find the reviewers' comments useful as you decide how to proceed. Should additional work allow you to address these criticisms, we would be happy to look at a substantially revised manuscript. If you choose to take up this option, please either highlight all changes in the manuscript text file, or provide a list of the changes to the manuscript with your responses to the reviewers.

When resubmitting, please provide a point-by-point response to the reviewers' comments. Please submit your responses as a separate file, distinct from your cover letter where you can add responses to the Editors' comments that you do not want to be made available to the reviewers. Word files are preferred. We recommend that any figures, tables or graphs that are included in the response to reviewers are also included in the main article or Supplementary Information.

If the revision process takes significantly longer than three months, we will be happy to reconsider your paper at a later date, as long as nothing similar has been accepted for publication at Communications Earth & Environment or published elsewhere in the meantime.

Please use the following link to submit your revised manuscript, point-by-point response to the reviewers' comments with a list of your changes to the manuscript text (which should be in a separate document to any cover letter), a tracked-changes version of the manuscript (as a PDF file) and any completed checklist:

Link Redacted

Please do not hesitate to contact us if you have any questions or would like to discuss the required revisions further. Thank you for the opportunity to review your work.

Best regards,

Alienor Lavergne, PhD
Senior Editor
ORCID: 0000-0002-4591-1217
Communications Earth & Environment

EDITORIAL POLICIES AND FORMAT

If you decide to resubmit your paper, please ensure that your manuscript complies with our editorial policies and complete and upload the checklist below as a Related Manuscript file type with the revised article:

Editorial Policy Policy requirements
(Download the link to your computer as a PDF.)

- Behavioural and social science
- Ecological, evolutionary & environmental sciences
- Life sciences

<https://www.nature.com/documents/nr-reporting-summary.zip>

For your information, you can find some guidance regarding format requirements summarized on the following checklist: (<https://www.nature.com/documents/commsj-phys-style-formatting-checklist-article.pdf>) and formatting guide (<https://www.nature.com/documents/commsj-phys-style-formatting-guide-accept.pdf>).

REVIEWER COMMENTS:

Reviewer #1 (Remarks to the Author):

Reviewer Report on "Welfare Implications of Heat Stress-Induced Labour Productivity Loss under Global Warming Scenarios in Burkina Faso"

Summary of the Paper: The authors analyze the economic impacts of heat stress-induced labor productivity losses in Burkina Faso, with a focus on heterogeneous effects across agroecological zones (AEZs). Using a country-specific static CGE model calibrated to a 2019 SAM, they simulate the economic consequences of projected increases in heat stress under two warming scenarios (+1.5°C and +3.5°C). The study finds significant sectoral disparities, with agriculture and mining suffering the largest losses. Furthermore, rural households, particularly in the Sudanian and Sahelian zones, bear the greatest welfare losses, while urban households experience mixed impacts due to shifting factor prices and consumer costs. The authors highlight potential policy measures to mitigate heat stress effects, including mechanization, agroforestry, and climate adaptation strategies.

Overall Assessment: This study contributes valuable insights into the economic effects of heat stress at a granular, subnational level, filling an important gap in the literature. The methodological approach, combining climate-induced labor productivity shocks with a detailed country CGE framework, is sound and well-motivated. The paper is well-structured and presents clear findings with important policy implications. However, there are several areas where improvements could enhance the clarity, robustness, and broader applicability of the study.

Major Comments

1. Limited Consideration of Global Spillover Effects

An important drawback of using a country CGE lies in that it assumes fixed international prices, meaning that e.g., global agricultural and mining markets elsewhere are unaffected by heat stress. In reality, widespread heat stress would likely cause global price shifts, potentially offsetting some domestic economic losses. The authors should consider how this could be addressed while staying within the country-specific framework. E.g., testing sensitivity to exogenous global price increases of the most heat-impacted sectors/commodities could provide a more nuanced understanding of how Burkina

Faso might be impacted within an interdependent global economy.

2. More Emphasis on Subnational Heterogeneity

A key strength of the study is its disaggregation across AEZs, but the discussion of regional variations, especially in the results section, is somewhat underdeveloped. Consider whether one of the supplementary figures that show considerable AEZ variation e.g. Fig S2 may be better placed in the main body. Also consider expanding the results discussion of how sectoral compositions and pre-existing vulnerabilities differ across zones. For instance, is agriculture concentrated in specific AEZs? Are there baseline economic differences that explain why some AEZs experience sharper declines?

3. Exploring Factor Substitution Sensitivity

The study tests sensitivity to labor-labor substitutability but does not explore labor-capital substitution. In reality, mechanization may serve as an adaptation strategy. An alternative sensitivity test increasing labor-capital substitutability could offer insights into how accelerated structural shifts toward capital-intensive production will impact the results shown

4. Clarifying the Drivers of Large Sectoral Impacts

The study finds that mining output drops by up to 71% under +3.5°C warming, a striking result that warrants further discussion. Does mining in Burkina Faso rely disproportionately on unskilled labor? Is there significant labor reallocation away from mining to other sectors? Does showing a decomposition of production changes by labor reallocation or showing employment shifts clarify these dynamics.

Specific Comments

1. References Formatting:

References within the text are incorrectly formatted (e.g., “ref.11” instead of author-year citations). This issue appears multiple times.

2. Results Section:

The size of the labor productivity shocks should be presented upfront. Consider including a summary of sectoral and regional labor productivity losses within the main text (currently only in the supplement).

Figure 2 (production effects) could be supplemented with an additional panel showing changes in labor employment by sector to clarify production shifts.

3. Household Welfare Analysis:

The positive welfare effect for urban poor households is surprising. Could this be further explained? Do urban households have very large consumption shares (of meat & fish) so that they are benefiting disproportionately from falling livestock and fishing prices?

The authors should clarify that the livestock and fisheries sectors are also vulnerable to heat stress affecting animals, which is yet to be explicitly modeled here and in CGE literature in general. In real-world price responses, livestock and fish products can also be expected to see price increases

4. Discussion section:

The discussion currently implies that the author’s results differ significantly from ref.13, but in reality, the strong mining effects in this paper are not contradictory—ref.13 already shows a ~6% decline in mining labor productivity, compared with ~10% in this paper. Perhaps the point to be made here instead is that by not accounting for sub-national variations, productivity and economic impacts can be under-estimated by non-trivial amounts

The discussion acknowledges that the GDP contraction is largely driven by the effects on export sectors such as cotton and mining. However, this raises an important question: How much of the contraction is due to falling domestic productivity versus how much is due to assumed fixed world prices? Would global price changes (higher agricultural and mineral prices due to heat stress elsewhere) mitigate some of the losses? Addressing this briefly, or at least acknowledging it as a limitation, would strengthen the interpretation.

The discussion mentions that urban poor and non-poor households gain welfare in the +1.5°C scenario. While the authors attribute this to lower consumer prices and higher skilled wages, this deserves more generalizable discussion. Does the improved welfare hinge on certain consumption patterns and price declines?

The limitations of the study are generally well stated, but a few points could be added: Capital-Labor Substitution: The study assumes a relatively fixed production structure with limited adaptation. Would more flexible labor-capital substitution affect results? International Trade Effects: The study does not account for changing global trade dynamics due to heat stress. Are there ways future work to better incorporate this while also maintaining sub-national heterogeneity and/or using a country-

specific CGE model?

5. Presentation of Figures:

Figure 3: The visual differences across AEZs are not clear. Consider adjusting the color gradient or adding a numerical summary.

Figure 5: The macroeconomic effects are substantial, but interpretation could be improved by linking specific sectoral declines to aggregate GDP changes.

Final Remarks This paper makes a valuable contribution to understanding the economic consequences of heat stress at a national and subnational scale. The methodological rigor is commendable, and the results are policy-relevant. Addressing the points raised above would enhance the robustness of the findings and broaden the study's applicability. Expanding sensitivity analysis, refining the discussion of subnational heterogeneity, and incorporating additional insights into sectoral dynamics would strengthen the paper further. I recommend some revisions before publication.

Reviewer #2 (Remarks to the Author):

Review: Welfare Implications of Heat Stress-Induced Labour Productivity Loss under Global Warming Scenarios in Burkina Faso

Country-specific analysis using a CGE model is commendable. However, the calibration of the model is rather poor.

1. The authors are using labour capacity and not labour productivity, this is a major issue as labour productivity is a measure of economic performance. Additionally, the economic impact of labour capacity loss is unclear.
2. WBGT-labour productivity/capacity response-functions are a major weakness in the literature studying climate impacts on labour and this paper does not contribute anything substantial to change that fact.
3. Recent literature has highlighted major weaknesses with WBGT (d'Ambrosio Alfano et al., 2014; Golbabaei et al., 2021; Kong and Huber, 2022; Vecellio et al., 2022; Dasgupta et al., 2024). The findings from this paper are likely affected. However, the authors do not discuss these issues. This is another major weakness.

References

- Dasgupta, S., Robinson, E.J.Z., Shayegh, S. et al. (2024). Heat stress and the labour force. *Nat Rev Earth Environ* 5, 859–872. <https://doi.org/10.1038/s43017-024-00606-1>.
- d'Ambrosio Alfano, F.R., Malchaire, J., Palella, B.I., Riccio, G. (2014). WBGT Index Revisited After 60 Years of Use, *The Annals of Occupational Hygiene*, 58(8);955–970, <https://doi.org/10.1093/annhyg/meu050>.
- Golbabaei, F., Ahmadi, Asour A., Keyvani, S., Kollahdoozi, M., Mohammadiyan, M., Fasih-Ramandi, F. (2021) The Limitations of WBGT Index for Application in Industries: A Systematic Review. *Int J Occup Hyg.*; 13(4);365-381.
- Kong, Q., and Huber, M. (2022). Explicit calculations of wet-bulb globe temperature compared with approximations and why it matters for labor productivity. *Earth's Future*, 10, <https://doi.org/10.1029/2021EF002334>.
- Vecellio, D.J., Tony, S. W., Cottle, R.M., and Kenney, W.L. (2022) Evaluating the 35°C wet-bulb temperature adaptability threshold for young, healthy subjects (PSU HEAT Project). *Journal of Applied Physiology*, 132:2, 340-345. <https://doi.org/10.1152/jappphysiol.00738.2021>.
- Somanathan, E., Somanathan, R., Sudarshan, A. and Tewari, M. (2021). The Impact of Temperature on Productivity and Labor Supply: Evidence from Indian Manufacturing. *Journal of Political Economy*, 129:6, 1797-1827. <https://doi.org/10.1086/713733>.

Communications Earth & Environment is committed to improving transparency in authorship. As part of our efforts in this direction, we are now requesting that all authors identified as 'corresponding author' create and link their Open Researcher and Contributor Identifier (ORCID) with their account on the Manuscript Tracking System prior to acceptance. ORCID helps the scientific community achieve unambiguous attribution of all scholarly contributions. You can create and link your ORCID from the home page of the Manuscript Tracking System by clicking on 'Modify my Springer Nature account' and following the instructions in the link below. Please also inform all co-authors that they can add their ORCIDs to their accounts and that they must do so prior to acceptance.

Version 1:

Decision Letter:

Dear Mr Houessou,

Your manuscript titled "Welfare Implications of Heat Stress-Induced Labour Productivity Loss under Global Warming Scenarios in Burkina Faso" has now been seen by our reviewers, whose comments appear below. In light of their advice we are delighted to say that we are happy, in principle, to publish a suitably revised version in Communications Earth & Environment.

We therefore invite you to revise your paper one last time to address the remaining concerns of our reviewers. At the same time we ask that you edit your manuscript to comply with our format requirements and to maximise the accessibility and therefore the impact of your work.

EDITORIAL REQUESTS:

****Please take care to match our formatting and policy requirements. We will check revised manuscript and return manuscripts that do not comply. Such requests will lead to delays. ****

SUBMISSION INFORMATION:

OPEN ACCESS:

Communications Earth & Environment is a fully open access journal. Articles are made freely accessible on publication. For further information about article processing charges, open access funding, and advice and support from Nature Portfolio, please visit <https://www.nature.com/commsenv/open-access>

Link Redacted

Best regards,

Alienor Lavergne, PhD
Senior Editor, Communications Earth & Environment
Consulting Editor, Communications Sustainability
ORCID: 0000-0002-4591-1217

Springer Nature
The Campus, 4 Crinan Street, London N1 9XW, UK
www.nature.com/commsenv
@commsearth.bsky.social

REVIEWERS' COMMENTS:

Reviewer #1 (Remarks to the Author):

The authors have made sufficient revisions in response to my previous comments. The manuscript is clear and with limitations clearly described. I have no further concerns and am happy to recommend it for acceptance in its current form.

** Visit Nature Portfolio's author and referees' website at www.nature.com/authors for information about policies, services and author benefits**

Welfare Implications of Heat Stress-Induced Labour Productivity Loss under Global Warming Scenarios in Burkina Faso

Manuscript ID COMMSENV-25-0275-T

Dear editor,

dear reviewers,

we would like to thank you for giving us the opportunity to submit a revised draft of our manuscript titled “Welfare Implications of Heat Stress-Induced Labour Productivity Loss under Global Warming Scenarios in Burkina Faso” to *Nature Communications Earth & Environment*. We appreciate the time and effort that has been put into providing us with valuable feedback and comments. Based on these comments, we have incorporated changes into the manuscript, which have been highlighted using the track changes feature. Below, we provide a detailed answer to all the comments and concerns raised by the reviewers.

Reviewer 1

Major comments

No.	Comment	Response
1	Limited Consideration of Global Spillover Effects – An important drawback of using a country CGE lies in that it assumes fixed international prices, meaning that e.g., global agricultural and mining markets elsewhere are unaffected by heat stress. In reality, widespread heat stress would likely cause global price shifts, potentially offsetting some domestic economic losses. The authors should consider how this could be addressed while staying within the country-specific framework. E.g., testing sensitivity to exogenous global price increases of the most heat-impacted sectors/commodities could provide a more nuanced understanding of how Burkina Faso might be impacted within an interdependent global economy.	Indeed, single-country CGE models, including the one used in our study, assume fixed international prices. We do not see this as a problem regarding the effects of a small country such as Burkina Faso on world markets, which are negligible. But indeed, if heat stress affects many countries, as it would do in reality, this is likely to cause global price shifts. To include such international price effects caused by heat stress, a sensitivity analysis could be conducted by changing world market prices in the model. To do this, we would require the results of global CGE models for world market price changes induced by heat stress under similar global warming scenarios as considered in our study. Such simulations are not available. We therefore highlight this limitation now in our discussion section. The text incorporated (see discussion section, page 13): Single-country CGE models, including the one applied in this study, typically operate under the assumption of fixed international prices. This implies that our analysis only accounts for the effects of labour heat stress within Burkina Faso, but not for world market price-mediated effects of heat stress in other countries. However, global warming is expected to affect labour productivity in many countries. This may shift international prices especially in heat stress sensitive sectors such as for example cotton and mining, adding to the effects of changes in domestic labour productivity. For example, a rise in world market prices, e.g. for cotton and mining commodities, could incentivize increased production in Burkina Faso. This response could partially offset the production declines observed under the assumption of fixed international prices, especially in export-intensive sectors. Higher global prices may also result in

		higher prices of imported intermediate goods, such as fertilizers and machinery. Sectors dependent on these inputs may therefore decline, potentially amplifying the overall negative impact on GDP. Analyzing such a scenario would require estimates of international price changes, ideally derived from a global CGE model under the same warming scenarios as in this study, which is currently not available.
2	More Emphasis on Subnational Heterogeneity – A key strength of the study is its disaggregation across AEZs, but the discussion of regional variations, especially in the results section, is somewhat underdeveloped. Consider whether one of the supplementary figures that show considerable AEZ variation e.g. Fig S2 may be better placed in the main body. Also consider expanding the results discussion of how sectoral compositions and pre-existing vulnerabilities differ across zones. For instance, is agriculture concentrated in specific AEZs? Are there baseline economic differences that explain why some AEZs experience sharper declines?	As suggested, we placed former Fig. S2 into the main body (now Fig. 3). We extensively discuss it in the results section, addressing the questions: 1) Is agriculture concentrated in specific AEZs? 2) Are there baseline economic differences that explain why some AEZs experience sharper declines? The text incorporated (see results section, page 5): Maize, rice, and tubers are also exclusively produced in the Sudanian and Sudano-Sahelian zones. The higher the heat stress level, the greater the adverse effects on these crops. The production of sorghum, millet, fonio, peanuts, legumes, fruits, and vegetables shows a similar pattern, being more affected in the Sudano-Sahelian than in the Sudanian zone. Some of these crops, including rice, peanut and sorghum, even grow in the Sudanian zone. Compared to the Sudano-Sahelian zone, these crops are less labour-intensive in the Sudanian zone, resulting in lower or negligible heat stress effects. This is also the case for sorghum and peanuts in the Sahelian zone, which are either unaffected or increasing in both scenarios.
3	Exploring Factor Substitution Sensitivity – The study tests sensitivity to labor-labor substitutability but does not explore labor-capital substitution. In reality, mechanization may serve as an adaptation strategy. An alternative sensitivity test increasing labor-capital substitutability could offer insights into how accelerated structural shifts toward capital-intensive production will impact the results shown.	As you suggest, we conducted an additional sensitivity analysis, which increases the initial labour-capital substitution elasticity by 50%. As expected, the effects of labour productivity loss induced by heat stress are slightly lower due to the higher substitutability of labour and capital across sectors. In the main body, we extended the sensitivity analysis to these new results. A summary is provided in Fig. 7. We adapted the methodology section accordingly. Text incorporated (see results section, under “Macroeconomic effects” sub-section, page 10) The results also deviate slightly, with higher labour-capital substitutability (increased by 50%). The effects of labour heat stress are lower, as less productive labour can be better compensated for due to the higher labour-capital substitutability (see Fig. 7). We also include a sentence under the method section to signal that we conducted a sensitivity analysis for increasing labour-capital elasticities of substitution. Text incorporated (see method section, under “Sensitivity analysis” sub-section, page 16). We conducted a third sensitivity analysis, which increased labour-capital substitutability by 50%.

4	Clarifying the Drivers of Large Sectoral Impacts – The study finds that mining output drops by up to 71% under +3.5°C warming, a striking result that warrants further discussion. Does mining in Burkina Faso rely disproportionately on unskilled labor? Is there significant labor reallocation away from mining to other sectors? Does showing a decomposition of production changes by labor reallocation or showing employment shifts clarify these dynamics?	In response to your questions, we provided “An overview of labour employment and income as depicted by the developed SAM”. This is the first sub-section under the results section. The overview indicates that the mining sector relies heavily on unskilled labour, accounting for nearly 98% of the total labour employed. Additionally, the unskilled labour in the mining sector represents 30% of all the unskilled labour available in the economy. We have also added the changes in labour demand across sectors to the supplementary information (Table S1), which shows that labour is reallocating from mining to other sectors. Text incorporated (see results section, page 2) In 2019, unskilled and semi-skilled workers received 53% and 8% of the total revenue paid to labour, respectively (see Table 1). While representing only 5% (physical units) of total employment²⁵, skilled workers earned as much as 39% of the total labour revenue. Unskilled workers dominate all sectors in terms of physical labour employment. Regarding the labour income share, they dominate all sectors except the service sector. In the services sector, the income and employment shares of skilled workers are higher than in other sectors. A summary is provided in Table 1 (page 2)
---	--	--

Specific comments

No.	Comment	Response
1	References Formatting – References within the text are incorrectly formatted (e.g., “ref.11” instead of author-year citations). This issue appears multiple times.	Thanks for highlighting this mistake, which we corrected throughout the document.
2	Results Section – a) The size of the labor productivity shocks should be presented upfront. Consider including a summary of sectoral and regional labor productivity losses within the main text (currently only in the supplement). b) Figure 2 (production effects) could be supplemented with an additional panel showing changes in labor employment by sector to clarify production shifts.	a) We have moved the labour productivity shocks into the main body, under the Results section. We also provided a short explanation of these shocks. Text incorporated (see Results section, page 2-3) Table 2 presents the estimated heat stress-induced labour productivity shocks introduced into the CGE model. Labour productivity shocks in agriculture are differentiated across agroecological zones. Agriculture encompasses crops, livestock, fishing, aquaculture, and forestry. Accordingly, agricultural labour productivity shocks apply to all these subsectors. Labour productivity shocks to other sectors apply at the national level. Regardless of the warming scenario, workers in the service sector who engage in light-intensity work are the least affected by heat stress (see Table 2 and Fig. S9). This is followed by the manufacturing sector, where workers engage in moderate-intensity work (see Table 2 and Fig. S9). The sectors most affected are agriculture, mining, and construction, which require heavy work. In agriculture, labour productivity loss is

		most substantial in the Sahelian, followed by the Sudano-Sahelian and Sudanian zones. The productivity of skilled workers is least affected across sectors, as they often work indoors and are protected from solar radiation during their work time. Each additional temperature increase due to global warming further reduces the productivity of all workers, as indicated by the estimates for the +3.5°C warming scenario. b) As suggested, we added Table S1 to the supplementary information, which presents changes in labour employment by sector.
3	Household Welfare Analysis – a) The positive welfare effect for urban poor households is surprising. Could this be further explained? Do urban households have very large consumption shares (of meat & fish) so that they are benefiting disproportionately from falling livestock and fishing prices? b) The authors should clarify that the livestock and fisheries sectors are also vulnerable to heat stress affecting animals, which is yet to be explicitly modeled here and in CGE literature in general. In real-world price responses, livestock and fish products can also be expected to see price increases	a) Thank you for bringing this surprising result to our attention. In response to this comment, we thoroughly evaluated our base situation database. This exercise enabled us to spot a mistake in the factor income distribution for households. We re-run the whole analysis with the improved SAM. Household welfare changes are somewhat different now (while most other results are still very similar), with all rural and urban poor households experiencing a decline in welfare in all scenarios, and rural and urban non-poor households gaining or maintaining stable welfare in the +1.5°C scenario. In the +3.5°C scenario, these households also experience welfare loss, but to a lesser extent than their poorer counterparts. Text incorporated (See Results section, page 8) Labour productivity loss induced by heat stress affects household groups differently (Fig. 5). Generally, the heterogeneity of welfare effects stems from differences in the composition of factor income and consumption, as well as differences in heat stress-induced labour productivity losses across agroecological zones (Fig. S7 and S9). Rural poor households lose welfare under both warming scenarios. This result is primarily due to reduced income from supplying land, unskilled labour, and semi-skilled labour (Fig. S3). The greater welfare loss among these households is due to their heavy dependence on these production factors, in contrast to capital and skilled labour. The welfare of urban poor households is similarly affected, but to a lesser extent than that of their rural poor counterparts. Their welfare loss is also driven by reduced income, mainly due to declining wages for unskilled and semi-skilled labour. Besides the income effect, both rural and urban poor households face higher prices for crop commodities, which dominate their expenditure pattern. Both rural and urban non-poor households are unaffected or experience welfare gains in the +1.5°C warming scenario, primarily driven by declining consumer prices (Fig. S5) for commodities such as livestock, fishing/aquaculture, and services, which collectively represent a substantial share of their consumption expenditures (Table S3). Compared to poor counterparts, rural and urban non-poor households have a lower expenditure share for food commodities (Table S3) and

		are therefore less affected by increasing crop commodity prices (as shown in Fig. S5). Household groups are affected differently across agroecological zones (Fig. 5). Rural and urban poor households in the Sahelian zone lose slightly more welfare than their counterparts in other agroecological zones across both warming scenarios. This is due to their factor income from unskilled and semi-skilled labour being relatively higher compared to their counterparts in other zones. Among the non-poor households, the Sahelian zone is the most affected, because both rural and urban non-poor hold a relatively higher income share from unskilled compared to skilled labour than their counterparts in other zones. The wages of unskilled and semi-skilled labour are more affected than those of skilled labour, which is reflected in the welfare changes of these households in the Sahelian zone. Changes in household income and expenditure are available in the supplementary information, Figs. S3 and S4, respectively. Household income and expenditure composition from the reference scenario are also available in the supplementary information Tables S2 and S3. b) Thank you for this comment. In this study, we did not model the direct effects of heat stress on animals. This, coupled with the current labour productivity shocks, would have a higher compounded effect, likely leading to an increase in livestock and fishery prices. We made this limitation more explicit now. Text incorporated (see Discussion section, page 12) We do not account for the direct effects of heat stress on plants^{37,38} and animals³⁹⁻⁴². Assessing the combined effect of heat stress on plants, animals, and humans would likely find more severe economic effects. In both scenarios, we found declining prices for livestock and fishing commodities. Direct heat stress effects on animals, which are not modelled in this study, could result in lower production and higher prices for these commodities than presented here, as well as more negative economy-wide effects.
4	Discussion section –a) The discussion currently implies that the author’s results differ significantly from ref.13, but in reality, the strong mining effects in this paper are not contradictory—ref.13 already shows a ~6% decline in mining labor productivity, compared with ~10% in this paper. Perhaps the point to be made here instead is that by not accounting for sub-national variations, productivity and economic impacts can be underestimated by non-trivial amounts b) The discussion acknowledges that the GDP contraction is largely driven	a) Thank you for this observation. In response to this observation, we clarify this in the Discussion section. The differences stem from the approach used in our study to implement the labour productivity shocks. Text modified (see Discussion section, page 11) Even if the warming target of the Paris Agreement is met, which is below 2°C, heat stress-induced labour productivity loss could still lead to a substantial economic downturn, resulting in a GDP loss of 9%. In a +3.5°C warming scenario, GDP would decline, representing 20% relative to Burkina Faso’s base economy in 2019. This is consistent with previous studies^{10-13,28}, which also report an adverse GDP effect due to heat stress-induced labour productivity loss. We find a GDP loss under +1.5°C, which is more than double the GDP loss reported by Saeed et al. (2022)¹³ for a +3°C warming scenario. The

	by the effects on export sectors such as cotton and mining. However, this raises an important question: How much of the contraction is due to falling domestic productivity versus how much is due to assumed fixed world prices? Would global price changes (higher agricultural and mineral prices due to heat stress elsewhere) mitigate some of the losses? Addressing this briefly, or at least acknowledging it as a limitation, would strengthen the interpretation. c) The discussion mentions that urban poor and non-poor households gain welfare in the +1.5°C scenario. While the authors attribute this to lower consumer prices and higher skilled wages, this deserves more generalizable discussion. Does the improved welfare hinge on certain consumption patterns and price declines? d) The limitations of the study are generally well stated, but a few points could be added: Capital-Labor Substitution: The study assumes a relatively fixed production structure with limited adaptation. Would more flexible labor-capital substitution affect results? International Trade Effects: The study does not account for changing global trade dynamics due to heat stress. Are there ways future work to better incorporate this while also maintaining sub-national heterogeneity and/or using a country-specific CGE model?	difference mainly stems from different labour productivity shocks simulated. Saeed et al. (2022)¹³ implemented a uniform labour productivity shock of nearly 17% in agriculture, 6% in mining and construction, and 4% in manufacturing and other sectors. They did not distinguish outdoor and indoor labour in a given sector, but assumed that some sectors are outdoor, while others are indoor, and that productivity loss was the same for skilled or unskilled workers in each sector. Contrary to Saeed et al. (2022)¹³, we differentiate workers by skill level, and assume unskilled and semi-skilled workers are outdoors and skilled workers are indoors across all sectors. This leads to the implementation of labour productivity shocks being heterogeneous across labour types, and stronger on average by sector. b) We acknowledge this more explicitly now as one of the limitations of our study. c) We have extended the Discussion section to provide further explanation. Text incorporated (see modified text for question 4a) d) We conducted an additional sensitivity analysis with higher labour-capital substitutability. Overall, the result patterns remain the same; we only observed some minor deviations, which are presented in Fig. 7. Based on this sensitivity analysis, we have extended the suggested policy implications regarding mechanization. Text incorporated (see Discussion section, page 13) The strong adverse effects on labour-intensive sectors, mostly cotton and mining, signal the need for diversification into less labour-intensive production systems. For example, investments in machinery could be made in the cotton sector, rendering it more mechanized by increasing the use of tractors and harvesters. More automation in the mining sector could help offset the negative effects of climate change-induced declines in labour productivity. As shown by the sensitivity analysis above, a shift from labour to more capital-intensive production could mitigate the overall negative effects of labour heat stress. Regarding the effects of international trade, we stated this as a limitation of our study. As presented above in our response to question 1 under “Major Comment”, we outlined how this can be implemented in the model while maintaining sub-national heterogeneity in a single-country model.
5	Presentation of Figures – a) Figure 3: The visual differences across AEZs are not clear. Consider adjusting the color gradient or adding a numerical summary. b) Figure 5: The macroeconomic	a) Thank you for this suggestion. We adjusted the colour gradient for former Fig. 3, now Fig. 4, in the manuscript (see Page 7). b) In response to your comment, we extended the interpretation of macroeconomic effects.

	effects are substantial, but interpretation could be improved by linking specific sectoral declines to aggregate GDP changes.	Text incorporated (see Results section: Macroeconomic effects sub-section, page 10) The GDP loss can primarily be attributed to the substantial decline in crop production (especially cotton), mining, and non-food manufacturing, which collectively contribute a significant share of the overall GDP.
--	---	---

Reviewer 2

No.	Comment	Response
1	The authors are using labour capacity and not labour productivity, this is a major issue as labour productivity is a measure of economic performance. Additionally, the economic impact of labour capacity loss is unclear.	Thank you for pointing out the difference between labour capacity and productivity. We agree that the ISO labour response function used in our paper is more accurately interpreted as a measure of labour capacity, as it aims to avoid overly high body temperatures and prevent heat injuries (i.e., the capacity to work safely). Following the reviewer’s suggestion, we repeat the whole analysis using an alternative labour response function developed from the “Hothap” project [1] and recently adopted in the Lancet Countdown report after slight modification [2]. This labour response function is based on field studies of rice harvesters in India [3] and gold mining workers in South Africa [4] and should be interpreted as a measure of labour productivity, reflecting economic performance. We found that the “Hothap” function generates lower labour productivity loss than the ISO function, probably because rice harvesters and gold mining workers are highly heat-acclimated and may tend to work beyond their safety capacity. However, the pattern of variation across different regions, sectors, and labour types remains the same regardless of which labour response function is used. Namely, our results are at least qualitatively robust to the choice of labour response functions. For the sake of space, we only present the macroeconomic effects from using the “Hothap” function: Text incorporated (see Results section: Macroeconomic effects sub-section, page 10) We also repeat the whole analysis using an alternative labour response function as derived from the “Hothap” project²⁶ and modified and adopted in the Lancet Countdown report²⁷. The results remain qualitatively the same as using the ISO labour response function. Here, we present the results for the economy-wide effect, noting that similar findings apply to other types of impact. The labour shock effects are lower when using the Lancet labour-response function (Lancet-LRF) (Fig. 7). This is because the ISO function is designed to protect the health and safety of workers worldwide. In contrast, the Lancet function is derived from real-world field studies on highly heat-acclimated labour in subtropical countries, where workers may tend to work beyond their safety capacity (see the

		Method). As a result, the Lancet function estimates lower productivity loss compared with the ISO function (Table S5). However, both functions generate similar patterns of macroeconomic effects across different aspects of the economy, suggesting that our results are robust to the choice of labour response functions. Text incorporated (see Method section, under “Sensitivity analysis” sub-section, page 16) Finally, we repeated the entire analysis using an alternative labour response function derived from the “Hothap” project, based on two empirical studies: one on gold mining workers in South Africa⁶³ and another on rice harvesters in India⁶⁴. The “Hothap” function was slightly modified and adopted in the Lancet annual countdown report²⁷, and the Lancet modified version (Eq. 3 in Saeed et al. (2022)¹³) is used in this work. We decided to keep the ISO function for most of the paper for two reasons. First, the “Hothap” function is derived from highly fit and heat-acclimated individuals with the motivation to work beyond their safety capacity, which may make it less applicable to other scenarios. Second, working beyond safety capacity may cause heat morbidity or even mortality, and therefore leads to productivity losses, which, however, are not accounted for in the “Hothap” labour response function.
2	WBGT-labour productivity/capacity response-functions are a major weakness in the literature studying climate impacts on labour and this paper does not contribute anything substantial to change that fact.	We completely agree that more empirical evidence is needed to refine the WBGT-labour productivity/capacity response functions, potentially through field studies and chamber experiments. It’s especially important to collect such evidence across different climate regimes and types of work. However, we think efforts along this line fall outside the scope of this study. Our novelty and main contribution lie in how we implement existing labour response functions to evaluate the economic impact of heat stress-induced labour loss in a warmer world. We explicitly account for the heterogeneity of working intensity and sun exposure profiles across different sectors, labour skills levels, and agroecological zones within Burkina Faso. We also explicitly assess the welfare impact on different types of households (poor and non-poor, as well as rural and urban). This allows us to provide more granular information on the heterogeneous impact of labour loss due to heat stress increase with climate change, which can better inform local adaptation strategies.

3	Recent literature has highlighted major weaknesses with WBGT (d'Ambrosio Alfano et al., 2014; Golbabaie et al., 2021; Kong and Huber, 2022; Vecellio et al., 2022; Dasgupta et al., 2024). The findings from this paper are likely affected. However, the authors do not discuss these issues. This is another major weakness.	Thank you for raising this issue. We completely agree that WBGT has limitations. For example, WBGT always decreases with wind speed, whereas a stronger wind can increase human heat strain when the air is much hotter than the skin [5]. WBGT was also found to underestimate heat stress under humid and low wind conditions [6]. In addition, WBGT is unable to account for the fact that as it gets hotter, our body becomes increasingly sensitive to humidity [6]. In fact, not only WBGT, but all heat stress metrics are faced with various limitations, as it is very challenging, if not impossible, for any single metric to capture the complexity of the human body and working scenarios. Among all heat stress metrics, WBGT is still considered the standard metric for determining working-rest cycles and is adopted by multiple standards for workplace heat regulation [7-9]. It has been found to perform as well as, or better than, other metrics in evaluating the physiological impact of heat exposure [10], modelling heat-related health risks such as heat stroke morbidity [11], and examining how multiple meteorological factors interact to influence human physical work capacity [5, 12]. However, it's unclear how the limitations of WBGT would affect the estimations of labour response especially at spatial and temporal aggregate level (such as annual labor loss for Burkina Faso) mainly due to two reasons. First, our application involves various types of environmental conditions (e.g., low or high wind, hot-dry or warm humid) and working scenarios (indoor or outdoor) and biases caused by WBGT limitations can have different signs under different conditions with the net impact being unclear. In addition, the WBGT limitations in quantifying heat stress can be absorbed to some extent when developing the WBGT-labor response function. As a result, we are unable to provide informative discussions on the impact of WBGT limitations on labour loss estimation. But we now mention this limitation in the discussion: Like all heat stress metrics, WBGT faces various limitations^{30,31}, which may lead to under- or overestimating labour productivity loss from heat stress. Still, it has been found to perform as well as, or better than, other metrics in evaluating the physiological impact of heat exposure^{32,33}.
---	---	--

Reference:

- [1] Kjellstrom, T., Freyberg, C., Lemke, B., Otto, M. & Briggs, D. Estimating population heat exposure and impacts on working people in conjunction with climate change. *Int J Biometeorol* 62, 291–306 (2018).
- [2] Watts, N. et al. The 2020 report of The Lancet Countdown on health and climate change: responding to converging crises. *The Lancet* 397, 129–170 (2021).
- [3] Sahu, S., Sett, M. & Kjellstrom, T. Heat Exposure, Cardiovascular Stress and Work Productivity in Rice Harvesters in India: Implications for a Climate Change Future. *Ind Health* 51, 424–431 (2013).
- [4] Wyndham, C. H. Adaptation to heat and cold. *Environmental Research* 2, 442–469 (1969).
- [5] Foster, J. et al. Quantifying the impact of heat on human physical work capacity; part II: the observed interaction of air velocity with temperature, humidity, sweat rate, and clothing is not captured by most heat stress indices. *Int J Biometeorol* (2021) doi:10.1007/s00484-021-02212-y.
- [6] Budd, G. M. Wet-bulb globe temperature (WBGT)—its history and its limitations. *Journal of Science and Medicine in Sport* 11, 20–32 (2008).
- [7] ISO (2017). Ergonomics of the thermal environment—Assessment of heat stress using the WBGT (wet bulb globe temperature) index (International Standard). International Organization for Standardization.
- [8] ACGIH (2017). 2017 tlvs and beis: Based on the documentation of the threshold limit values for chemical substances and physical agents and biological exposure indices (Tech. Rep.). ACGIH.
- [9] NIOSH (2016). Criteria for a recommended standard: Occupational exposure to heat and hot environments (Tech. Rep. No. DHHS (NIOSH) publication No. 2016-106). DHHS, NIOSH.
- [10] Ioannou, L. G. et al. Indicators to assess physiological heat strain – Part 3: Multi-country field evaluation and consensus recommendations. *Temperature* 9, 274–291 (2022).
- [11] Guo, Q. et al. Comparing Integrated Heat Stress Indicators With Raw Meteorological Variables in Predicting Heat Stroke-Related Ambulance Transportations in Japan.
- [12] Havenith, G., Smallcombe, J. W., Hodder, S., Jay, O. & Foster, J. Comparing efficacy of different climate indices for predicting labor loss, body temperature, and thermal perception in a wide variety of warm and hot climates. *Journal of Applied Physiology* jappphysiol.00613.2023 (2024) doi:10.1152/jappphysiol.00613.2023.